# Barriers and facilitators to adherence to secondary stroke prevention medications after stroke: analysis of survivors and caregivers views from an online stroke forum

James Jamison,[1] Stephen Sutton,[1] Jonathan Mant,[1] Anna De Simoni[2]

[1]Primary Care Unit, Department of Public Health and Primary Care, Forvie Site, University of Cambridge School of Clinical Medicine, Cambridge, UK
[2]Centre for Primary Care and Public Health, Barts and the London School of Medicine and Dentistry, Queen Mary University of London, London, UK

**Correspondence to**
James Jamison; jj285@medschl.cam.ac.uk

## ABSTRACT

**Objective** To identify barriers and facilitators of medication adherence in patients with stroke along with their caregivers.

**Design** Qualitative thematic analysis of posts about secondary prevention medications, informed by Perceptions and Practicalities Approach.

**Setting** Posts written by the UK stroke survivors and their family members taking part in the online forum of the Stroke Association, between 2004 and 2011.

**Participants** 84 participants: 49 stroke survivors, 33 caregivers, 2 not stated, identified using the keywords 'taking medication', 'pills', 'size', 'side-effects', 'routine', 'blister' as well as secondary prevention medication terms.

**Results** Perceptions reducing the motivation to adhere included dealing with medication side effects, questioning doctors' prescribing practices and negative publicity about medications, especially in regard to statins. Caregivers faced difficulties with ensuring medications were taken while respecting the patient's decisions not to take tablets. They struggled in their role as advocates of patient's needs with healthcare professionals. Not experiencing side effects, attributing importance to medications, positive personal experiences of taking tablets and obtaining modification of treatment to manage side effects were facilitators of adherence. Key practical barriers included difficulties with swallowing tablets, dealing with the burden of treatment and drug cost. Using medication storage devices, following routines and getting help with medications from caregivers were important facilitators of adherence.

**Conclusions** An online stroke forum is a novel and valuable resource to investigate use of secondary prevention medications. Analysis of this forum highlighted significant barriers and facilitators of medication adherence faced by stroke survivors and their caregivers. Addressing perceptual and practical barriers highlighted here can inform the development of future interventions aimed at improving adherence to secondary prevention medication after stroke.

## INTRODUCTION

Stroke recurrences are associated with higher disability and mortality than first time strokes,[1]

### Strengths and limitations of this study

► The first study to explore and identify perceived barriers and facilitators to medication adherence among users of an online stroke forum using a theoretical framework.

► Inclusion of caregivers offers a unique perspective for understanding of medication taking behaviour in patients with severe disabilities.

► The online forum allowed patients and caregivers to communicate in a comfortable environment beyond the influence of the research team, shedding new light on factors affecting adherence to secondary prevention medications.

► Key themes identified may be limited by the search terms used and may not provide an exhaustive list of all barriers and facilitators to adherence discussed in the forum.

► Posts were scrutinised by a moderator prior to being published.

► The online forum was dated 2004–2011 and might not reflect current practice.

and account for one-third of all stroke events.[2] Secondary prevention medicines are considered important in reducing stroke recurrences in patients who have already experienced a stroke or transient ischaemic attack.[3][4] However, medication non-adherence is an issue known to be problematic, contributing to suboptimal health outcomes.[5] Reported practical barriers to medication adherence after stroke include forgetting medication, difficulty swallowing tablets and difficulties handling packaged medications.[6] A key factor decreasing stroke patients' motivation for taking secondary prevention medications is having concerns about tablets, such as becoming dependent on them or worrying about their long-term effects.[7] Difficulties with taking medication, lack of information on stroke and medications and

patient's fears of medicines are important barriers, while support from caregivers and worrying about further stroke are facilitators.[8]

Severe stroke-related impairments make it difficult for survivors to participate in research. Perhaps for this reason there is little evidence available on factors affecting adherence to medications in patients with more disabling strokes (at least a third of stroke survivors).[9 10] Studying factors affecting adherence can be difficult because of self-presentational bias, that is, patients may perceive that a certain behaviour, for example, adherence to treatment, is one of the duties expected of the 'good patient' and may be reluctant to admit a different behaviour, or reactivity bias, that is, if patients are aware that their adherence is being monitored, this might increase adherence simply by drawing attention to it.[11] In a recent investigation, De Simoni *et al* used an online forum to explore adherence to inhaler treatment in asthma adolescents according to a framework, gaining fresh insights on factors affecting adherence in this patient group.[12]

Our analysis differs from previous adherence literature by assessing survivors and caregivers attitudes to medication adherence from a viewpoint that has not been previously explored. The online forum offers users the opportunity to discuss issues around medication that may be considered sensitive and which they may be less willing to address through traditional face-to-face approaches.

Caregivers of elderly patients experience difficulties with tablet administration.[13] As patients' dependency on caregivers for medicine taking increases, caregivers' factors also become relevant in determining patients' adherence.[10] Indeed, a recent interview study by the authors[14] highlighted the importance of caregivers in adherence to secondary prevention medications. Findings from another study suggest that among patients with cardiovascular disease, those with a caregiver were more likely to be adherent to medications.[15] There is a need to investigate adherence to secondary prevention medications after stroke avoiding self-presentational and reactivity biases, including patients with severe disabilities and caregivers factors.[10] Online health forums are accessible 24/7 in the form of asynchronous communication that is convenient to the user. This medium offers anonymity and encourages honesty. Individuals with health-related difficulties can communicate in confidence about what matters to them.[11] TalkStroke is an online forum where survivors and their families discuss information and provide support to one another. Recent investigations using this forum showed that a wide variety of themes were discussed online, including secondary prevention medications.[16 17] Caregiver views were well represented and most of them (70%) looked after patients with severe disabilities. Among forum users who were stroke survivors, 30% were suffering from severe impairments.[16] Therefore, data from the archives of this forum have the potential to shed light on adherence issues for these hard-to-reach groups.[18 17]

The aim of this investigation was to understand barriers and facilitators of medication adherence among survivors of stroke and their caregivers through evaluating posts written in an online stroke forum, using a framework-based approach.

## METHODS
### Design
We undertook a qualitative analysis of posts to the TalkStroke online forum using the Perceptions And Practicalities Approach (PAPA) theoretical framework.[18] According to the framework, non-adherence is viewed as a variable that can change over time and treatments. Non-adherence is known to be intentional or unintentional. Unintentional adherence is linked to practical factors and resource limitation, for example, forgetting to take medications because of lack of prompting or experiencing difficulties with swallowing tablets. Perceptual factors or beliefs affect intentional adherence, that is, how patients consciously make decisions that influence their medication taking behaviour.[19] This occurs when patients deliberately choose not to follow recommendations and where beliefs about medications influence motivation to start and continue treatment. The PAPA framework was chosen as it is specifically designed to identify and classify factors affecting adherence to medications. Results have the potential to inform the development of behavioural interventions aimed at improving adherence and their subsequent evaluation according to causal pathways. The framework posits that patients make a choice to take medication based on judgement of their personal need for the medication, relative to their concerns about the possible consequences of taking it.[18] The PAPA approach seeks to understand adherence through addressing both perceptual (beliefs and preferences) and practical (capability and resources) factors which have an influence on patients commencing and continuing treatment. We searched the forum archives using a set of predefined keywords, in order to identify barriers and facilitators of adherence to secondary prevention medications. Posts were written by stroke survivors or family members/caregivers.

### Setting
The analysis was performed on the archives from TalkStroke, a UK-based online forum hosted by the Stroke Association website, including 22 173 posts written between 2004 and 2011 by 2583 unique usernames.[16] TalkStroke is an online resource through which stroke survivors and caregivers could seek and/or offer information and support. Forum users could discuss any topics, develop their own conversation threads and there was no restriction on the subject discussed. Participants could read the subject of the thread being discussed and decide whether they wished to contribute. Differentiating survivors and caregivers was done by reading the text of the post: survivors talked in first person about themselves,

**Table 1** Key themes highlighting survivors' and caregivers' barriers and facilitators to adherence to secondary prevention medications classified according to perceptions and practicalities.

| Sample characteristics | N | Median range |
|---|---|---|
| Total participants identified in posts | 84 | |
| Number of posts in the forum/ participant | | 16 (1–4932) |
| Number of posts about secondary prevention medications/ participant | | 1 (1–37) |
| Age at stroke | | |
| Survivor | | 50 (32–72) |
| Patient by caregiver | | 66 (46–91) |
| Gender | | |
| Male—survivor | 20 | |
| Female—survivor | 26 | |
| Not known—survivor | 3 | |
| Male—patient talked about by caregiver | 20 | |
| Female—patient talked about by caregiver | 12 | |
| Unknown gender and unknown identity | 3 | |
| Identity person posting | | |
| Stroke survivor | 49 | |
| Caregiver | 33 | |
| Not known | 2 | |
| Years since stroke | | |
| (0–12 months) | 37 | |
| (1–5 years) | 25 | |
| (6–10 years) | 4 | |
| (11–15 years) | 2 | |
| (15+ years) | 1 | |
| Unknown | 15 | |
| Caregiver identity | | |
| Daughter/son | 20 | |
| Spouse | 9 | |
| Other (/in-law/ sister) | 3 | |
| Unknown | 1 | |

while caregivers were talking about a stroke survivor in the third person, for example, "my father had a stroke". Stroke survivors with severe disabilities were among the users of the forum. Caregivers could register as users independently from patients; 60% of users were in fact caregivers.[16] We acknowledge that some caregivers could have assisted patients in writing their posts, although we do not have data to quantify these occurrences.

### Procedure and participants

A word list of unique terms of the archive file of Talk-Stroke was generated using AntConc3.2.4.[20] Terms related to secondary prevention medications were selected (eg, amlodipine, statin, warfarin, ramipril), including misspellings (eg, asprin, simvastin), brand names (eg, Lipitor, Plavix) and drug categories (eg, statin, diuretics, blood pressure medicines, etc). Posts including any secondary prevention medication term were identified. We additionally searched the TalkStroke archive for the keywords: 'taking medication', 'pills', 'size', 'statins', 'side effects', 'capsule', 'box', 'routine', 'blister' and 'secondary prevention'. These keywords were lay terms used by patients with stroke and their caregivers when talking about adherence to secondary prevention medications as emerged from the transcripts of a previous interview study by the authors.[14] The aim of the interviews was exploring stroke survivors' and caregivers' views around barriers and facilitators of adherence to secondary prevention medications in general practice. Search terms were discussed and agreed by two authors (JJ and ADS) (see online supplementary file 1).

Participants of the online forum included stroke survivors and patients talked about by caregivers, identified by usernames linked to each of the selected posts. Characteristics were retrieved from usernames, taking advantage of data from a previous study.[16] Demographics are shown in table 1. All posts that were relevant for the research questions were copied and pasted into Microsoft Excel and NVivo V.10 for later analysis.

### Ethics

The Stroke Association gave permission to ADS to use the data for research purposes. To protect the identity and intellectual property of forum participants,[21] we chose not to use *verbatim* quotes, despite this being normal practice in qualitative research. Instead, we used descriptions of quotes throughout the text.[16 22] Paraphrasing of the text reflected as closely as possible the original posts and was agreed among authors to minimise interpretation bias. The ethical aspects of conducting research on this forum have been discussed more extensively elsewhere.[16]

### Data analysis

A qualitative approach using thematic analysis was undertaken to explore forum posts.[23] All posts retrieved through the search terms were read by JJ to aid familiarisation. To strengthen the validity of findings and ensure rigour, 50% of all posts were double-coded by ADS. Throughout the process, the authors checked the coding structure obtained to ensure a high level of agreement in coding was maintained. Once completed, coding was compared and the intercoder reliability was measured. The kappa score was 80%. Queries arising from the coding process were resolved through discussions involving a third author (SS) where necessary, until a final consensus was reached. NVivo V.10 was used to manage and organise the data. A set of codes representing key themes were initially developed from the forum posts by JJ to represent barriers and facilitators of medication adherence. These themes were refined, and subthemes were

identified and grouped together with similar concepts. A coding framework was formed and refined further as additional themes emerged. Data saturation was reached with the recruitment of 84 individuals, beyond which no new themes emerged. Guided by the PAPA framework, we coded forum posts to identify practical and perceptual factors affecting adherence to medications. Identified themes were mapped onto the theory and subdivided into barriers or facilitators of adherence.

While we were unable to ask questions to clarify themes, users could participate in forum discussions they were interested in, offering insights on barriers and facilitators to adherence that may be beyond the reach of interviews. A previous investigation comparing an online forum with qualitative interviews concluded that the forum could provide useful data for qualitative health research.[24]

Although there is evidence that inappropriate medical information or health behaviours in this online stroke forum were identified and corrected by participants in subsequent postings,[16] consistent with what is reported elsewhere in a cancer patients' internet support group,[25] threads of discussions were not analysed here. Therefore, self-correction received through the forum is not reported. For the purpose of this study, the term caregiver refers to family members such as spouses or children, and is not associated with paid caregivers. The online supplementary file 1 shows the analysis pathway to reach the final number of themes.

## RESULTS
### Identification of posts
A search of the stroke forum resulted in 19 214 posts not associated with medication taking being excluded, leaving 2959 posts (see online supplementary file 1). Additional analysis excluded 2527 posts not about secondary prevention medications. Of the 473 remaining posts, a further 251 were removed as being duplicate or not directly associated with medication adherence. This yielded a total of 222 posts describing barriers/facilitators of adherence to secondary prevention medication, including 162 posts by stroke survivors, 57 by caregivers and 3 by an individual whose status as a survivor or caregiver could not be identified.

### Characteristics of study participants
From 222 posts related to adherence to secondary prevention medications, we identified 84 individuals. Approximately 60% of participants were stroke survivors posting about their own experiences with the remainder being caregivers, predominantly sons or daughters. The age of participants ranged from 32 to 91 years, male and female were similarly represented (n=40 vs 38; table 1). The majority of participants experienced a stroke within 12 months of posting on the forum. Around three-quarters of participants (73%) reported a stroke occurring within the last 5 years, with 44% having had a stroke within the previous 12 months. The mean number of

years since stroke was 2 years 2 months for survivors and 10 months for patients talked about by caregivers. Several participants were prolific users on the forum and were instrumental in facilitating discussions and providing feedback on a considerable number of topics, offering a rich and in-depth assessment of issues raised. One forum participant wrote 37 posts about secondary prevention medications, while another 15 posts. The majority of participants (n=44) posted only once, 19 participants twice and 6 three times. Sample characteristics are reported in table 1.

### Themes
The range of themes comprising barriers and facilitators of medication adherence are reported in table 2. In line with the PAPA approach,[18] these are discussed according to the following two categories.

#### Perceptions—necessity beliefs and concerns
In this section, perceptual barriers and facilitators of medication adherence in stroke survivors and caregivers are explored, according to their classifications as necessity beliefs, that is, doubts about personal need for medication to maintain or improve current and future health, and their concerns about secondary prevention treatment.

#### Practicalities—capability and resources
In this section, we explore barriers and facilitators that stroke survivors and caregivers face around their capability of taking/giving medication and the resources available to undertake such behaviour.

Within each category, themes are grouped into barriers and facilitators. For each of the emerging themes, when relevant, caregivers' views have been reported after patients' ones.

## PERCEPTIONS OF SECONDARY PREVENTION MEDICATIONS
### Necessity beliefs and concerns

#### Treatment necessity

#### Lack of perceived benefits of medications
Doubts about high cholesterol as risk factor for stroke
A few users expressed doubts about the role of cholesterol in stroke, questioning the need for taking any preventative medications at all.

*A male survivor acknowledged statins controlled cholesterol, but believed strokes occurred regardless of cholesterol levels. He talked about the 'Cholesterol Myth' having researched the topic online and described feeling confused about taking statins when in reality they weren't needed. (Male, aged 67, age at stroke 55, N.70)*

#### Doubts about the added benefit of statins
Doubts were also expressed about the benefits statin added to long-term health outcomes.

**Table 2** Characteristics of online TalkStroke participants as identified in the study posts

| Perceptions | |
| --- | --- |
| **Barriers** | **Facilitators** |
| **Necessity beliefs** | **Necessity beliefs** |
| **Lack of perceived benefits of medication** | **Attributing importance to medications** |
| ▲ Questioning the effectiveness of secondary prevention medications in preventing stroke recurrence. | ▲ Recognising taking tablets as important to prevent stroke recurrence. |
| ▲ Considering statins detrimental to health and not effective. | ▲ Feeling reassured by taking secondary prevention medications. |
| ▲ Valuing adherence but recognising that it is the choice of the patient to take tablets. | ▲ Experiencing consequences of non-adherence (a further stroke) as driver of necessity beliefs. |
| ▲ Realising that stroke could still occur despite taking secondary prevention medications. | ▲ Feeling secondary prevention medications are important and should only be stopped in consultation with the GP. |
| | ▲ Holding strong beliefs about the need for secondary prevention medications. |
| **Concerns** | **Concerns** |
| **Management of medication side effects*** | **Management of medication side effects*** |
| ▲ Experiencing statins side effects and considering they have potential to worsen quality of life. | ▲ Awareness that not all patients are affected by side effects. |
| ▲ Changing diet/lifestyle as alternative to taking medication to reduce side effects.* | ▲ Healthcare professionals changing medications to counteract side effects. |
| ▲ Healthcare professionals recommending diet and exercise to reduce cholesterol instead of taking statins.* | ▲ Modifying medications to achieve optimal treatment. |
| ▲ Struggling to raise issues about side effects of statins with healthcare professionals and obtaining changes in treatment when patients find it unsuitable. | ▲ Obtaining changes in treatment from healthcare professionals until side effects are manageable. |
| **Impact of bad press on statins** | **Trusting healthcare professionals** |
| ▲ Being extracautious about commencing statins for fear of side effects. | ▲ Perceiving medications to be beneficial as secondary healthcare professional also taking it. |
| ▲ Struggling to ensure patients' adherence to statins in face of bad press. | |
| **Questioning prescribing practices** | |
| ▲ Being disappointed as medications considered ineffective were not changed by GPs. | |
| ▲ Having concerns around incorrect medications being prescribed. | |
| ▲ Receiving conflicting information about medications. | |
| ▲ Worrying about medications being prescribed for financial reasons or guidelines over clinical judgement. | |
| ▲ Experiencing difficulties with asking GPs to prescribe alternative tablets as current ones considered unsuitable. | |
| ▲ Feeling the role of GPs is limited to advising about secondary prevention medications. | |
| ▲ Intentionally missing medications to manage side effects. | |
| **Practicalities** | **Practicalities** |
| **Barriers** | |

Continued

| Table 2 Continued | |
|---|---|
| **Problems associated with taking tablets**<br>▲ Swallowing medication capsules, especially big size ones.<br>▲ Experiencing difficulties with handling medications due to size and stroke-related physical impairments.<br><br>▲ Experiencing frustration with burden of multiple medications and episodic patients' refusal of medications.<br>▲ Experiencing frustration at patient refusal to take statins and attend routine medication appointment.<br>▲ Experiencing difficulties when helping patients with aphasia taking tablets in the context of changes in treatment.<br>▲ Experiencing frustration at patients failing to keep up with refilling prescriptions.<br>**Cost of medications**<br>▲ Struggling to meet the costs of secondary prevention medications. | **Storage devices for managing medication**<br>▲ Using pill box: helping seeing the correct medication was taken and when prescription needed to be renewed.<br><br>▲ Using pill boxes to provide written instruction to patients or keeping a note of tablets taken.<br>▲ Advice from pharmacist on taking medication correctly.<br>**Good medication taking routines**<br>▲ Linking tablet use to an everyday activity to facilitate medication taking behaviour.<br><br>▲ Assuming control of medication when patients have problems with short-term memory and reminding when tablets have to be taken. |

*Because of missing details of the underlying clinical scenario, these themes could act both as barriers or facilitators to adherence to secondary prevention medications, therefore, have been reported under both headings.
Statements in italics refer to caregivers' themes.
GP, general practitioner.

*A female survivor read about the hype around statins and stated she still didn't have confidence in them. She had read a research paper on statins suggesting they only added an extra 9 months of life.* (Female, aged 56, age at stroke 56, N.66)

### Caregivers' related views

### Respecting patients' medication choice
Caregivers struggled with their role of ensuring patients' adherence. They felt survivors' decisions about choosing or refusing medications needed to be evaluated according to patients' preferences and not just in terms of what was clinically right.

*A caregiver recognized it was hard to encourage her father to take medications. He suffered many side effects which made him feel less in control so he would choose to go without tablets. She said it was important to have a balance regarding what the survivor wanted, considering he stated he would be happier if he felt he was in control. See concluded that patient's choice had to be respected, even if she didn't agree.* (Male, age unknown, age at stroke unknown, N.46)

### Awareness of stroke recurrences despite medications
The fact that survivors could suffer a further stroke despite taking secondary prevention medications and following a healthy lifestyle also raised concerns around the benefits of adherence to medications.

*A caregiver described how after having a first stroke, her father changed his lifestyle completely by eating well, exercising more and taking medication to control his blood pressure. However 1 day his BP surged suddenly and he experienced a second stroke.* (Male, age unknown, time since stroke 0 years, N.55)

### Attributing importance to medications
### Secondary prevention medications are essential to prevent stroke recurrences
The importance of secondary prevention medication in reducing the risk of a stroke event was acknowledged by forum users. Prioritising secondary prevention tablets over other types of medications highlighted the significance survivors attached to adherence to these medications. These posts were often written in reply to users complaining of medication side effects.

*A female survivor commented that it was better to take a few extra tablets from the GP than to experience another stroke. Tablets were provided to prevent a further stroke, and she stressed that they shouldn't be stopped except on professional advice.* (Female, aged 51, age at stroke 51, N.17)

### Secondary prevention medications offer reassurance
Another survivor reported feeling reassured by medications, particularly warfarin.

*A female survivor mentioned that although she had suffered 2 strokes in the previous year, none had occurred since commencing warfarin. She felt reassured about taking warfarin and she was now worried about coming off the medication as she had already experienced flashing in her left eye since she had started to be weaned off the drug.* (Survivor, female, aged 42, age at stroke 42, N.35)

### Experiencing the consequences of non-adherence improves adherence

Experiencing the consequences of medication non-adherence after having another stroke reinforced necessity beliefs about secondary prevention medications.

*A survivor who had already suffered 2 strokes acknowledged it was impossible to ever fully recover from the stroke experience. He said after his first stroke he was prescribed tablets he didn't take and after suffering the second stroke he realised this was a big mistake.* (Male, aged 67, age at stroke 55, N.82)

*A survivor refused statins after her first stroke because of side effects. However, after suffering a second one she was now worried enough to take them.* (Survivor, female, aged 68, age at stroke 67, N.14)

### Cargivers' related views

### Not taking secondary prevention medications is risky

Caregivers generally held strong beliefs about the need for secondary prevention medications.

*A caregiver (husband) advised that if patients don't take medications they're likely to become worse. He was amazed about how many people choose not to take their tablets, perhaps half of them, and few even did so when they knew they had a meeting with the consultant in the coming weeks.* (Female, aged 46, age at stroke 46, N.12)

*A caregiver (daughter) mentioned that her father wasn't taking medication routinely. He had had a massive stroke just a few weeks earlier. She wanted to say to forum users that if stroke survivors follow a healthy lifestyle and are strict with medications, then there is no reason why a major stroke could not be prevented.* (Male, aged 55, age at stroke 55, N.6)

## CONCERNS
### Management of medication side effects
#### Suffering from side effects contributes to suboptimal adherence

The experience of side effects led some users to intentionally alter adherence to the medications. This was done by 'making a compromise' with healthcare professionals.

*A male survivor described being suspicious of the number and variety of pills he was dispensed. He said that he had come to a compromise with his doctor about taking blood pressure tablets. He was on 2 tablets for blood pressure, of which one was a diuretic. Having got fed up of frequently running to the toilet, he decided to check his blood pressure every day and would skip the diuretic if blood pressure was*

*fine.* (Male, age unknown, age at stroke unknown, N.63)

### Lifestyle changes versus taking secondary prevention medications

To avoid side effects, some stroke survivors took the decision to reduce cholesterol through changing diet, rather than medications, without mentioning whether this decision was taken with or communicated to healthcare professionals. However, reducing cholesterol through diet rather than medication was recommended by the GP also.

*A female survivor decided to reduce her cholesterol through diet because of unpleasant side effects of statins. Once symptoms disappeared, she wouldn't take the statins, but instead olive oil and a healthy diet to keep her cholesterol balanced naturally. She said she would continue aspirin as it didn't seem to cause side effects.* (Female, aged 52, age at stroke 52, N.76)

*A female survivor mentioned her cholesterol level was average. Her nurse suggested starting medication but her GP was against this, saying the level could be reduced through diet and exercise alone as these tablets were over prescribed. She added that statins were recommended when needed because of genetic makeup (meaning familial hypercholesterolemia).* (Female, aged 49, age at stroke 48, N.21).

Depending on the exact clinical scenario, the decisions about statins in the last two posts could be medically appropriate or not, that is, act both as barrier or facilitator to adherence to secondary prevention medications. Due to lack of details, no definite classification could be made. To reflect this, themes were reported under both headings in table 2, but reported only here within the results, for simplicity.

### Caregivers' related views

### Caregiver difficulties as advocates of stroke survivors with healthcare professionals

Caregivers assumed at times the role of advocates for their family members suffering from the side effects of medication and reported struggling in this role. Failure to be successful in obtaining a change in treatment led some survivors to stop taking medication completely.

*A female caregiver described consistently trying to have her husband's 40 mg statin dosage reduced by his GP. As a result of the high dosage he was chronically tired, so he stopped taking statins.* (Male, aged 54, age at stroke 52, N.68)

### Impact of bad press on statin
#### Influence of side effects on taking medicines

Side effects of secondary prevention medications raised important concern, and statins were frequently discussed by forum users. The bad press about statins was mentioned in relation to starting the medication and ongoing adherence. Participants discussed these concerns together with healthcare professionals.

*A survivor wrote that despite her GP's recommendation she couldn't commence statins after reading in the press about side effects. She said she felt well and didn't want to jeopardise that, as she wasn't convinced she needed them. Although also her consultant disagreed with her decision and was keen for her to take them, he said she didn't necessarily have to take them. (Female, aged 54, age at stroke 54, N.37)*

### Caregivers' related views

### Bad press making harder for caregivers to encourage adherence
Reading information about statins and their side effects highlighted caregivers' struggle and made it more difficult for them to help stroke survivors be adherent.

*A caregiver's mother had suffered 2 mini strokes and was now prescribed both aspirin as well as pills to lower cholesterol but was refusing to take these as she had read in the press about the bad side effects they caused. (Female, age unknown, age at stroke unknown, N.74)*

### Questioning prescribing practices
#### Problems with obtaining appropriate secondary prevention medication treatment
Disappointment was expressed when practitioners failed to start/change secondary prevention medications when the survivor judged their current treatment to be inadequate.

*A survivor described feeling let down as he requested changes in medications because he didn't feel they (aspirin and clopidogrel) were beneficial. He'd lost confidence in the health care system after visiting several consultants and being sent home with unchanged medications. (Male, aged 43, age at stroke 41, N.20)*

### Concerns around incorrect prescribing
This was also apparent when the prescribed medication was perceived as being incorrect.

*A stroke survivor recalled being on 75 mg of aspirin as well as beta blockers, however, his nephew who was a consultant surgeon, suggested that had he been taking warfarin instead of the aspirin he may not have suffered a second stroke. (Male, aged 67, age at stroke 55, N.82)*

### Inconsistent advice about medications prescribed
Receiving conflicting advice on medication practices caused further uncertainty and confusion, which might have indirectly affected adherence to secondary prevention medications.

*A survivor suffered increased bleeding while on warfarin was taken off it. He suffered another stroke shortly after, and was put back on warfarin for the bleeding to begin again. He felt confused at being told to stay on warfarin to avoid a potentially serious stroke. (Male, aged 72, age at stroke 72, N.10)*

### Caregivers' related views

### Questioning GP's motivation to prescribe
Caregivers too raised concerns about GPs prescribing, principally statins, for financial rather than medical reasons, which could indirectly affect adherence, especially in patients suffering from statin side effects.

*A caregiver (sister) suggested that GPs shouldn't be paid for prescribing statins and that the decision should be based on clinical judgement alone. She suggested medication could be overprescribed as a result for financial reasons. (Gender and age unknown, age at stroke unknown, N.78)*

### Caregivers' difficulties as advocates of patients' medications
The caregivers' role as advocates for their family members came up in questioning prescribing practices, highlighting caregivers' awareness of guidelines and difficulties at times with obtaining treatment modifications on the behalf of patients. (The cost of atorvastatin has dropped since, so this post does not reflect current practice.)

*A caregiver recommended being firm with GPs about being put on atorvastatin if simvastatin was not tolerated, as atorvastatin was a bit more expensive but recommended by NICE guidelines as an alternative. (Gender and age unknown, age at stroke unknown, N.18)*

### GPs' role advising about secondary prevention medications
Some survivors reflected on the role of GPs in their adherence. They felt that the GP's role was to provide advice. Getting support from family in medication related decisions was considered important.

*A male survivor agreed to stop taking a blood pressure tablet with his doctor because of intolerable side effects, and his wife being a nurse made it easier. He felt strongly that doctors are there to advise not instruct. (Male, age unknown, age at stroke unknown, N.63)*

### Caregivers' related views
Caregivers also recognised the importance of medications and the need to continue taking tablets despite experiencing side effects. The importance of only stopping medication on GP's advice was highlighted.

*A caregiver reported that because of side-effects her husband had voluntarily come off all the medication he was taking, except for aspirin which he continued to use. She said they had agreed to this together with the GP and stressed the importance of doing so before stopping tablets. (Male, aged 54, age at stroke 52, N.68)*

### Management of medication side effects
#### Medications did not necessarily cause side effects
Survivors who did not experience medication side effects generally felt that taking medication was a positive preventative measure against stroke. Although threads

of discussion were not analysed, these posts often were written in reply to users who complained about suffering from side effects.

*A male survivor advised it was better taking tablets than risking another mini-stroke. He had a severe stroke himself and was prescribed aspirin and simvastatin. He never experienced side effects and also knew others on the same statin who didn't experience any either. (Male aged 67, age at stroke 63, N.52)*

### Changing medications to avoid side effects

Forum users reported changes in secondary prevention medications being made by the health professionals to counteract negative side effects, which helped adherence.

*A male survivor described that on a dosage of 8 mg of warfarin he started to suffer migraines and bleeding, leading him to refuse the drug. After further conclusive tests, the consultant decided to take him off warfarin as he was taking persantin, which never gave him a headache or nosebleed. He acknowledged warfarin was an important drug, but didn't suit everyone. (Male, aged 49, age at stroke 49, N.47)*

### Perseverance with asking about modifications to achieve optimal treatment

Doctors' and patients' perseverance in modifying medications was important to achieve optimal treatment.

*A male survivor reported taking up to 7 different blood pressure tablets and that it was unusual for a stroke patient to only need a few. He recommended going back to the GP as necessary to keep changing tablets until the right combination was found. (Male, aged 52, age at stroke 52, N.64)*

### Caregivers' related views

### Treatment adjustments to avoid side effects

Reduction of medication dosage by doctors and elimination of side effects was reported as a successful strategy to aid adherence.

*A female caregiver described her husband suffering from considerable side effects from simvastatin 40 mg but when the GP changed to atorvastatin at a lower dose of 10 mg he was able to cope. (Male, aged 54, age at stroke 54, N.49)*

## Trusting healthcare professionals

Healthcare professionals had an important role in patients' trust in secondary prevention medications and consequently adherence.

*A survivor described how he trusted his vascular surgeon who had changed his medication from warfarin to aspirin and statin. The survivor was happy to take aspirin and felt it would be good to continue as the surgeon also took it regularly, concluding it must be beneficial. (Survivor, male, aged 35, age at stroke 34, N.71)*

## PRACTICALITIES OF SECONDARY PREVENTION MEDICATIONS
## Capability and resources

### Problems associated with taking tablets
### Swallowing and handling medicines

Swallowing difficulties were reported when taking tablets, especially in relation to the medication dipyridamole, due to its size.

*A male survivor described 'swallow panic', that is, fear of choking when trying to take Dipyridamole capsules. The user reported it took around 3 months before he got over that. (Male, aged 67, age at stroke 55, N.70)*

Size of tablets also caused handling difficulties due to stroke-related impairments.

A survivor agreed with another user about the problem with the size of dipyridamole tablets, which were getting stuck in the pill box organizer. (Female, aged 46, age at stroke 45, N.30)

### Caregivers' related views

### Treatment burden

Taking multiple tablets also contributed to treatment burden experienced by caregivers. One caregiver described how this added to the survivor's episodic refusal to take any medications.

*A caregiver was asking advice on encouraging medication taking. He said his mother was on multiple tablets, up to 4 times a day, but was now refusing to take any at all and this did upset him. Persuading her to continue taking the most important tablets had taken hours to do. (Male, aged 77, age at stroke 77, N.9)*

### Attending routine appointments

Another practical difficulty was dealing with routine appointments which were considered burdensome, resulting in the survivors being non-adherent to medications.

*A caregiver (wife) described how her husband was adamant that he was not prepared to take statins because he didn't have the time to keep going back to the GP for check-ups. The caregiver reported feeling helpless. (Male, aged 55, age at stroke 55, N.24)*

### Difficulties experienced by patients with disabilities

Caregivers of patients with severe disabilities such as aphasia and inability to communicate, made their job of ensuring patients' adherence a difficult experience.

*A caregiver said she couldn't imagine what a stroke survivor was going through, with her mother unable to communicate following a stroke. She described her mother having difficulties with medications caused by previous changes in treatment. She felt her mother was giving up and wanted*

*advice on dealing with aphasia.* (Caregiver-daughter, aged 52, age at stroke 47, N.54)

### Problems with using storage devices

Using dosette boxes was sometimes a struggle for survivors with severe disabilities, and a source of worry for caregivers.

*A caregiver mentioned that despite using a nomad tray, tablets were still being taken from the wrong day with several days' worth of tablets being taken in a single day. His father in law often didn't take the time to work out the days or to look at the calendar.* (Male, age unknown, age at stroke unknown, N.40)

### Seeking advice from pharmacists on managing medications

Another caregiver described having to seek advice on the best way to manage the stroke survivor's medications.

*A caregiver said he went to the pharmacist and spent half an hour chatting about medications after which he bought a flip top multi-coloured medication box labeled with the days and doses. He also said it took him a while to establish the best way to fill the box without getting confused, eventually filling it a tablet at a time across the entire week, instead of a day at a time.* (Male, aged 82, age at stroke 82, N. 57)

### Cost of medication

Survivors' highlighted difficulties faced with meeting the cost of stroke medications.

*A female survivor described being prescribed both aspirin and simvastatin that she had to pay for. She reported having to take out a credit card to pay for her medications as she was unable to work and did not have any money coming or any benefits.* (Female, aged 59, age at stroke 59, N.72)

### Storage devices and strategies for medication management
#### Using medication aids

Stroke survivors also reported benefits from using medication aids including pill boxes and medication wallets to facilitate medication taking behaviour. These devices ensured the appropriate medication was being taken at the right time, while also allowing monitoring when boxes needed to be refilled.

*A survivor agreed the storage box was useful to view medication and her husband didn't have to keep asking her whether she had taken her tablets as he could also see. She said it was irritating to be constantly asked. The box helped her also with not running out of medications as she filled it weekly and could tell when it was time for a repeat prescription.* (Female, aged 46, age at stroke 45, N.30)

#### Caregivers' related views

#### Using medication instructions

Caregivers highlighted how instructions were considered helpful in facilitating day-to-day medicine taking. Keeping

track of medicines that had been taken was suggested as a method of ensuring good adherence.

*A caregiver (son) described making a note on the pill box asking the survivor to turn it over after taking the pills as this would mean the morning pills were now taken. A second instruction invited the survivor to do the same when taking the evening tablet. He suggested to forum users that a simple chart tracking when each medication was taken was also helpful.* (Caregiver-son, aged 82, age at stroke 82, N.57)

### Good medication taking routines
#### Creating good medication routines

Linking daily tablet use to an everyday activity or placing tablets in a specific location which then acted as a cue to take the medication was described as helpful by several users.

*A survivor suggested using a white board and having method in place helped. She remembered taking her own medications through repetition or linking tablet use to another everyday activity.* (Female, aged 54, age at stroke 46, N.19)

#### Caregivers' related views

#### Reminding survivors about taking tablets

Caregivers also played a key role in medication routines when survivors could not remember to take tablets.

*A caregiver (wife) described regularly giving her husband his medication because stroke had caused short term memory loss and he would forget them or sometimes take them over again. She said she was now in total control of his medications which was fine because she was a nurse with experience of this.* (Female, aged 46, age at stroke 40, N.5)

### DISCUSSION
#### Summary of main findings

Data from an online forum provided a rich source of information, illuminating on practical and perceptual barriers and facilitators to adherence to secondary prevention medications in stroke survivors and their caregivers. These data highlight several points. Concerns around the bad press on statins could result in stroke survivors being cautious about commencing/keep taking the medication, and opting for a change in diet as an alternative (potentially not a medically appropriate decision and without healthcare professionals' support). Survivors expressed concerns about being prescribed medications they considered inappropriate, questioned GPs' motivation to prescribe medications and at times realised when prescribing mistakes occurred. Caregivers themselves reported some doubts about the effectiveness of tablets and difficulties in ensuring good medication adherence, while recognising that it is ultimately a survivor's decision whether or not to take medication, particularly when suffering from side effects. Indeed, not experiencing

side effects from secondary prevention medications was an important facilitator of adherence. Health professionals successfully modifying treatment to manage side effects and awareness that not everyone suffers from side effects were reported as increasing the motivation to take secondary prevention treatment. Believing that medications reduced stroke risk, feeling reassured by taking secondary prevention treatment and experiencing another cerebrovascular event as consequence of non-adherence were important drivers of necessity beliefs and supported adherence.

Practical barriers included difficulties swallowing capsules, burden of multiple medications, stroke-related communication impairments (eg, aphasia) causing patients' confusion with any treatment changes, difficulties meeting medication costs and with managing storage devices. Caregivers' posts greatly contributed to these data. They reported that improved patients' adherence was linked to using medications storage devices, getting help from pharmacists in organising medicines, assuming full control of their family members' medication taking and having previous experience and knowledge about medications and their administration.

## Strengths and limitations

This study has a number of strengths. First, the method of data collection where descriptions by forum users capture unprompted thoughts is unlikely to be affected by self-presentational bias. Information comes from patients over a wide geographical area and includes people who might not take part in traditional research because of severe disabilities, communication impairments or in the case of caregivers, because of lack of time.[11] The forum creates a natural environment facilitating exchange in opinions and in-depth discussions around several topics including secondary prevention medications. The important presence of caregivers in online discussions is a further strength, offering a unique viewpoint on medication taking behaviour of survivors with severe disabilities. Given that patients with significant disabilities may not traditionally participate in health research, the online forum may represent a potentially important method of data collection through which these patients' views may be heard through their caregivers.

These findings however should be interpreted with caution. A key limitation of this research was that forum data were from the years 2004–2011 and therefore the findings reported here may not reflect current practice in primary care. Lack of details about the underlying clinical scenarios described in some of the posts made it difficult classifying emerging themes as barriers or facilitators to adherence. In addition, barriers and facilitators were limited to those identified from the predefined search criteria. Different keywords may have uncovered additional barriers to medication adherence we failed to identify, or revealed issues related to medications in general rather than specifically secondary prevention ones. All forum posts were examined by a moderator prior

to being published online, which may have restricted the views of some users. Finally, with the majority of forum users under the age of 70 years, it is possible that this method of data collection overlooks a significant proportion of the older stroke population.

## Comparisons with existing literature

Our investigation shed light on the significance stroke survivors and caregivers attributed to the bad press on statins, which impacted on their adherence. This is in agreement with a recent investigation concluding that negative statin-related news stories was associated with early discontinuation of statin and increased risk of death by cardiovascular disease.[26] Furthermore, people already taking statin were found to be more likely to stop this medication following high media coverage,[27] or when side effects were not tolerable despite GP's attempts to modify treatment.[17] Beliefs about secondary prevention medications differed at times between survivors and caregivers. Some stroke survivors decided to stop medications because of intolerable side effects, despite their caregivers' believing optimal adherence was important to prevent stroke recurrences. In the context of medication side effects, caregivers believed in their role as patients' advocates with healthcare professionals (including GPs and pharmacist) and often discussed and sought advice from other users in the forum on the matter. Findings from the present study also highlight the difficulties experienced by stroke survivors using blister packaged medication and dosette boxes, despite at the same time outlining their benefit in terms of adherence. Evidence from a systematic review has demonstrated a positive effect on adherence for those in the group using reminder packaging[28] as well as using pill boxes and blister packs in packaging interventions in cardiovascular disease,[29] while the use of reminder packaging may be a simple way of improving adherence to medication.[30] With older people known to experience difficulties taking medication, developing interventions that seek to combine the use of medication management devices with caregiver cooperation may be one way of addressing the practical challenges they face.

This study highlights a couple of interesting findings. Survivors reported making decisions about taking or not secondary prevention medications sometimes independently from their GPs, despite considering GPs' support important. Collaborative decision making involving caregivers, clinicians or pharmacists may however empower stroke survivors to make better informed decisions about secondary prevention medications. Understanding how patients make decision about medications is important[31] and GPs may benefit from enhancing caregivers' role in the decision-making process about medications.

Barriers to caring for the stroke survivor posthospital discharge have included a lack of collaboration with the healthcare team and a lack of community support for the caregiving role[32] as well as insufficient knowledge and skills to care for the survivor in the home.[33] We described in this study the struggle caregivers face in

their role as advocate of patients, on one side engaging with healthcare professionals for ensuring that recommended secondary prevention treatment is received, and on the other side wanting to support and respect patients' decisions about taking or not taking medications. Caregivers facing this dilemma could benefit from greater support by GPs and pharmacists. Caregivers could play an important role in bridging the gap between healthcare professional and stroke survivor in primary care and deserve more research and clinical attention. Developing interventions that seek to encourage active caregivers' engagement in stroke survivors' and healthcare professionals' shared decision making, can help to address more comprehensively barriers to adherence as well as delivering a care programme tailored to the individual needs of patients.[8]

Barriers highlighted here are in line with those reported by another qualitative study, where negative or erroneous beliefs about tablets, doubts around the effectiveness of medication and concerns about the consequences of not taking tablets were associated with being low adherers.[34] Greater emphasis on informing stroke survivors and caregivers about secondary prevention medications in primary care is needed. In a recent randomised trial evaluating an educational package for stroke survivors and caregivers, participants who received tailored information along with verbal reinforcement reported a greater satisfaction with medical and practical services.[35]

Survivors' concerns around the need for secondary prevention medications may reflect a wider pattern of misunderstanding about the benefits of such drugs. In an assessment of attitudes towards taking cardiovascular medications, caution expressed around medications was linked with how great the risk to health was perceived to be, with most patients saying they would do what the GP recommended.[36] A meta-analysis examining the necessity-concerns framework across a range of conditions found that experiencing the consequences of non-adherence reinforced the subsequent need to take tablets, acting as a driver of medication adherence,[37] in agreement with what is reported in this study by both stroke survivors and caregivers.

Although statins are known to reduce the risk of stroke by as much as 25%,[38] benefits are undermined by suboptimal adherence. In a previous examination on patient perspectives around statin therapy, compliance with statins was associated with information provided during the practitioner consultation as well as the beliefs about cholesterol and current health status.[39] This concurs with the findings in our study. In a recent investigation exploring non-adherence and patient's perceptions towards statins, it was found that almost three-quarters of all participants doubted the necessity of statins and lacked knowledge about this medication while concerns around side effects were significantly associated with intentional non-adherence.[40]

In this online forum, we found evidence that stroke survivors establish routines and use cues to facilitate medication taking. This is in agreement with previous findings from a pilot trial in which a plan to establish a medication taking routine resulted in significantly greater adherence among survivors.[41] Providing support to establish medication taking routines particularly among older patients with stroke can be beneficial.[42] Challenges to adherence with warfarin therapy, including beliefs about the need for this treatment have been highlighted previously, suggesting the benefit of a more collaborative patient-practitioner approach, focusing on education around anticoagulant therapy.[43]

These findings add to current literature by providing an assessment of adherence from users of an online forum. There has been little research on this approach to data collection conducted to date. The study identifies adherence concerns of a younger stroke population who may be less likely to be represented in research studies and whose attitudes to medication may be less well known. Findings add to the literature and shed lights on dynamic interactions between the survivor, caregiver and healthcare professionals and the extent to which this influences medication adherence in this patient group.

## Implications for clinical practice

Results of this investigation demonstrate the need to address identified barriers to adherence to secondary prevention medications within clinical practice. Improving patient-practitioner and caregiver-practitioner communication through more effective clinical consultations has the potential to benefit patients and encourage a greater understanding of the importance of secondary prevention medications. This approach could contribute to shaping patients' beliefs about medications and to improving confidence around taking them. Challenging negative medication beliefs, better information provision and addressing practical problems around the appropriate use of tablet storage devices, particularly for those patients with more severe disabilities as a result of stroke, has the potential to increase adherence and ultimately improve health outcomes.[40]

Both primary and secondary healthcare professionals should seek to engage the family of survivors and their support network to challenge concerns around taking tablets, offer reassurance on the benefits of medications, discuss the need for treatment in light of side effects and even support patients' informed decision to refuse medications.

Interventions using 'expert patients' or 'expert caregivers' providing support to stroke survivors and caregivers in the primary care setting hold potential.[15]

Internet fora for patients with stroke provide a potentially important resource through which the attitudes of survivors and their caregivers towards medication use can be better understood.

These findings provide new insight to clinicians about younger stroke survivors' concerns and the struggles caregivers might face in their role as patients' advocates. Awareness of these factors will improve consultations

about secondary prevention medication with both younger survivors and stroke survivors' caregivers. Stroke survivors with severe disabilities and their caregivers experience significant practical barriers to adherence. Greater focus on such practicalities by healthcare professionals would be beneficial.

This study highlights caregivers' unique position in overseeing patients' medications. Exploring the stroke survivor-caregiver dynamic can shed light on potential barriers to adherence to secondary prevention medication and ways to address them, eventually improving patients' outcomes.

## Future research

Our study suggests that caregivers play an important role in bridging the gap between patient and practitioner with regard to informing and facilitating the medication-taking process. Future research should therefore further explore their role in stroke survivors' medication taking and systematically incorporate them into adherence interventions.

Given the strong focus of forum users on statins, understanding why stroke survivors choose not to take statins as prescribed and suggesting to healthcare professionals effective ways of dealing with this issue should be a key focus for research in this area. With adverse events the most common reason for poor adherence to statin therapy, improved patient understanding of this medication through greater communication with the practitioner can help to address ongoing concerns.[44]

Future interventions should aim at further improving medication taking routines after stroke, using cues to prompt tablet taking. Advances in technology could facilitate delivery of such interventions. One novel approach to improving adherence particularly with regard to multiple medications is the use of fixed-dose combination therapy 'polypill approach'.[45] Indeed, a recent systematic review of barriers and facilitators of adherence to secondary prevention medications within cardiovascular disease found fixed dose combination (FDC) therapy to be an important facilitator associated with high adherence.[46]

## CONCLUSION

This study identified barriers and facilitators to medication adherence for stroke through analysing data from an online forum using a framework approach. Developing interventions which build on these results according to the framework has the potential to improve medication adherence and ultimately reduce the burden of stroke. Greater efforts are needed to meet the growing challenges faced by stroke survivors and their caregivers and to enable primary care clinicians to effectively address the burden of non-adherence to secondary prevention medications.

**Acknowledgements** The authors are grateful to the Stroke Association for permission to analyse the archives of the TalkStroke forum.

**Contributors** ADS conceived of the study, is the Chief Investigator, contributed to the data analysis and commented on the manuscript. JM is a co-investigator on the study, wrote and commented on the manuscript. SS is a co-investigator on the study, wrote and commented on the manuscript. JJ contributed to the study design, conducted the data analysis and prepared the manuscript for submission. All authors agreed on the final draft of the submitted manuscript.

**Funding** DS is funded by a National Institute for Health Research (NIHR) Academic Clinical Lectureship. The views expressed are those of the author(s) and not necessarily those of the National Health Service, the NIHR or the Department of Health. JJ was supported by a research grant from The Stroke Association and the British Heart Foundation.

**Competing interests** None declared.

**Provenance and peer review** Not commissioned; externally peer reviewed.

**Data sharing statement** No additional data are available.

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
