## [Reviewer comments · BMJ Open]

ARTICLE DETAILS

TITLE (PROVISIONAL)	Barriers and facilitators to adherence to secondary stroke prevention medications after stroke: Analysis of survivors' and caregivers' views from an online stroke forum
AUTHORS	JAMISON, JAMES; Sutton, Stephen; Mant, Jonathan; De Simoni, Anna

VERSION 1 - REVIEW

REVIEWER	Dr Elizabeth Taylor Kings College London UK
REVIEW RETURNED	11-Apr-2017

GENERAL COMMENTS	In this study, the authors analysed comments on an online forum in order to investigate barriers and facilitators to adherence to secondary stroke prevention medication. This is a useful and appropriate method for investigating this topic. In the main it is well-presented, but some revisions will make it easier to follow and less repetitive. Major comments: 1. The presentation of results could be organised in a way that is clearer and more concise. It is counter-intuitive that data reported under 'concerns' are sometimes actually the absence of a concern, or a facilitating factor. I wonder if there is a more straightforward way of using the PAPA framework to present the findings. For example, the framework distinguishes between intentional factors (motivation, perceptions) and unintentional factors (ability, practical factors), and it may be better to adhere more closely to this structure. A diagram could also be used to show visually how findings fitted into these categories. Attempting to divide the data into barriers and facilitators seems to cause problems as the authors attempt to force the data into one or the other category, rather than allowing that the same thing could be either a barrier or facilitator in different circumstances (see 3c below). At present there is repetition of findings under different headings, and the content does not always appear to be relevant to the heading. Investing some more time in re-structuring the findings would also enable a clearer focus on the new and important information the study has generated. Currently this gets rather lost. 2. I was concerned about the way that quotes were paraphrased as the wording used sometimes appeared to introduce bias. For example, the word 'admitted' is used frequently, which suggests that the participant was doing something wrong. (e.g. p19 "Another caregiver admitted having to seek advice on the best way to manage the stroke survivor's medication". 'Said', 'wrote', 'discussed', 'described' or 'reported' would be more objective verbs to use. I
---

	would like to see more exploration of the decision to paraphrase rather than use direct quotes in the ethics section. This only needs to be one or two sentences, but something to show that authors have considered and explained the ethical pros and cons. Did they tell users on the forum about the study? Did they ask them for an opinion on being quoted or para-phrased? Any sort of member-checking? If not, why not? 3. The discussion shows that the findings are in line with previous findings, but there should be more focus on the new and interesting findings in both the results section and the discussion. Interesting findings that I think are currently hidden: a) There are various comments on decision-making with or without health providers. This is interesting and I think would be a useful discussion point in relation to the literature on self-management, and I think there could be more focus on the contributions this study makes to this topic. For example, on p18 authors note that non-adherence sometimes resulted from patients not having time to keep going back to the GP for check-ups. There is also mention of the role of pharmacists in giving advice. Currently this is categorised under 'barriers' on p19, and this does not seem a good fit. If authors structured the findings as suggested above, this would just come under 'practical factors' and authors would have flexibility for a more nuanced explanation of how such factors might influence adherence. This could then be taken up in the discussion with consideration of how patients can be supported and empowered to make informed decisions. b) data on patients with communication impairments and their caregivers (p 18-19). This is an interesting finding and shows the benefit of using the chosen method of data gathering. c) Strategies for medication management. Currently this section beginning on p20 seems muddled. The heading 'storage devices for medication management' should be 'storage devices and strategies for medication management', as authors include data that is not about storage devices (e.g. making a note, using a chart). At the start of this section authors state that storage devices ensured the appropriate medication was taken, but this contradicts data presented in the previous section. Again, if this was presented under 'practical factors' it would be possible to discuss the fact that storage devices and strategies may or may not be helpful. Minor comments:  1. Discussion of the theory is lacking. Some critical discussion of why it was chosen is needed. On p5 the theory is explained but this could be re-worded for clarity, especially the sentence ending 'as well as unintentional adherence' around line 44. Linked with major comment 1) above, it is not clear how the theory has influenced the authors have presented the data. 2. p8 lines 19-23. This is a bit confusing. Did the majority of participants have a stroke within 12 months of posting, or post on the forum within 12 months of having a stroke? I would also suggest using years and months for time since stroke rather than e.g. 0.8 years. 3. p13 second quote needs re-wording - currently unclear 4. Use of apostrophes needs correcting throughout. This is my reason for stating that the standard of English is not currently suitable for publication in this journal.
--	--

REVIEWER	Nathalie Moise Columbia University Medical Center, USA
REVIEW RETURNED	14-Apr-2017

GENERAL COMMENTS

Jamison et al. provide an interesting qualitative analysis of factors affecting medication adherence amongst stroke survivors and their caregivers using data from an online health forum. Strengths include inclusion of caregivers as well as disabling patients, both under-assessed in the post-stroke adherence literature, use of online forums reducing self-serving biases and reducing bias introduced by focus group facilitators and a theoretical framework for data analyses. However, the article methodology, which relies on a query of an existing forum without using interview guides/repeat interviews, is a key flaw and may provide a biased viewpoint, though the authors reference 1 prior article that used a similar methodology published in this journal.

Introduction:

1. The authors provide a nice overview of the literature and gaps in viewpoints of patients with debilitating strokes. The authors might strengthen their argument by positing why/how their assessment of barriers and facilitators of adherence might differ from prior literature other than reducing "self-presentational bias", which would affect all adherence literature.
2. The authors seem to interchangeably use self-"presentational" and "representation" in the introduction.
3. While using an anonymous online forum is a strength, authors should include whether any prior research has used this methodology in assessing barriers/facilitators to adherence, specifically.

Methods

4. The authors should provide an argument for using this framework specifically as opposed to any others
5. How were keywords predefined (literature search?). How were authors assured that these would provide barriers/facilitators to adherence, specifically?
6. It would be helpful to better understand how TalkStroke is utilized and why. Do all caregivers use this forum and for what purpose? Can they discuss any topic or are there themes or subgroups that guide discussions? Given that many stroke patients had debilitating symptoms, how were authors able to clearly distinguish between the patient and caregiver (who themselves might assist the patient in posting on the forum?).
7. While open ended online forums allow for analyses of unprompted concerns around adherence, there are some methodological flaws that exist without interview guides and pre-specified questions/themes (e.g., identifying concerns related to adherence, specifically). Searching for "blister" provides very different implications from specifically elucidating (even on an online forum) barriers/facilitators to adherence in stroke patients, and contributes to measurement/coding bias. This method does not allow the authors to adhere to many of important COREQ items. How did the authors seek to mitigate this flaw? Is there literature on performing

rigorous qualitative online forum analyses?

8. The authors should provide a rationale for double coding 50% of the posts

9. The authors should provide a kappa score

Results

The authors provide comprehensive data on characteristics/demographics of the participants, despite the fact that this is an online forum. I found the data and results to be quite rich and interesting.

10. It would be helpful to quantify the number of individuals in each subgroup/theme.

11. The results were long and confusing. It would be helpful to better organize the results [e.g., changing “concerns” to “necessity concerns”; adding survivor headings to caregiver headings, placing all barriers together followed by facilitators, using similar fonts to specify themes, using more subheadings to differentiate different points, streamlining/shortening the caregiver results particularly since exact quotes are not utilized. It might be helpful to choose 1-2 examples of each subthemes (e.g., on pgs 10-11 there are too many examples of necessity concerns for “burden and management of medication side effects” unless some of these should be sub-themes)]

12. Unclear how authors were able to attribute these factors as adherence concerns. Was there high agreement among authors? Were posts excluded if not specifically pertaining to affecting adherence and how as this decided?

13. It's unclear why the authors chose to focus on adherence to statin/aspirin as opposed to also focusing on blood pressure agents for the purposes of secondary prevention.

14. Not all of the caregiver/survivor themes seemed appropriate under the same headings (“difficulty taking medications” has different implications than “difficulty administering medications” as an overburdened caregiver)

Discussion

Very nice overview of results and comprehensive review of limitations.

15. Minor edit on page 23: “older people know(n)”

16. While the authors highlight the ways in which their findings are in line with prior literature, it would be helpful to note the ways in which their findings also add to the literature. Did they find any differences from prior literature?

17. The authors might also highlight the ways in which caregiver/survivor beliefs differed, if at all.

18. While the future research section is strong, it would be helpful to better elucidate the ways in which these findings improve implications (“implications for clinical research” section), as their proposed implications are similar to prior literature.

VERSION 1 – AUTHOR RESPONSE

Reviewer: 1

Reviewer Name: Dr Elizabeth Taylor

Institution and Country: Kings College London, UK Competing Interests: None declared

In this study, the authors analysed comments on an online forum in order to investigate barriers and facilitators to adherence to secondary stroke prevention medication. This is a useful and appropriate method for investigating this topic. In the main it is well-presented, but some revisions will make it easier to follow and less repetitive.

Major comments:

1. The presentation of results could be organised in a way that is clearer and more concise.

It is counter-intuitive that data reported under 'concerns' are sometimes actually the absence of a concern, or a facilitating factor. I wonder if there is a more straightforward way of using the PAPA framework to present the findings. For example, the framework distinguishes between intentional factors (motivation, perceptions) and unintentional factors (ability, practical factors), and it may be better to adhere more closely to this structure. A diagram could also be used to show visually how findings fitted into these categories. Attempting to divide the data into barriers and facilitators seems to cause problems as the authors attempt to force the data into one or the other category, rather than allowing that the same thing could be either a barrier or facilitator in different circumstances (see 3c below). At present there is repetition of findings under different headings, and the content does not always appear to be relevant to the heading. Investing some more time in re-structuring the findings would also enable a clearer focus on the new and important information the study has generated. Currently this gets rather lost.

Response: The authors thank the reviewer for this suggestion, which has greatly improved readability of the results section. We have now more closely aligned the results to the structure of the PAPA framework as suggested. We have grouped the themes under the main headings Perceptions and Practicalities, removing the classifications into barriers and facilitators. We have added an additional sub-theme heading before each theme description to provide greater clarification on what each theme stands for.

2. I was concerned about the way that quotes were paraphrased as the wording used sometimes appeared to introduce bias. For example, the word 'admitted' is used frequently, which suggests that the participant was doing something wrong. (e.g. p19 "Another caregiver admitted having to seek advice on the best way to manage the stroke survivor's medication". 'Said', 'wrote', 'discussed', 'described' or 'reported' would be more objective verbs to use. I would like to see more exploration of the decision to paraphrase rather than use direct quotes in the ethics section. This only needs to be one or two sentences, but something to show that authors have considered and explained the ethical pros and cons.

Response: The authors thank the reviewer for highlighting this point, and agree the verb 'admitted' could appear as judgemental, which was not intended to. We have therefore replaced it with more objective verbs. See page 11, line 266; page 12, line 299; page 12, line 305; page 12, line 313; page 13, line 338; page 14, line 351; page 15, line 382; page 18, line 448; page 19, line 472; page 20, line 497; page 22, line 545.

We have added a sentence in the ethic section of the methods explaining that 'Paraphrasing of the text reflected as closely as possible the original posts and was agreed amongst authors to minimise interpretation bias'. See page 7, lines 185-186

Did they tell users on the forum about the study? Did they ask them for an opinion on being quoted or para-phrased? Any sort of member-checking? If not, why not?

Response: Given the number of participants and timespan of the data (2004-2011), contacting each single participant to request consent would have been impractical. We consulted with ethics experts and researched the literature about ethical issues in qualitative research on internet communities.

According to Eysenbach & Till (2001) <http://www.bmj.com/content/323/7321/1103.long> , we considered intrusiveness, potential for harm and perception of forum as public or private as key issues to consider regarding the ethical use of internet data. We came to the conclusion that paraphrasing quotes was the best way to protect the identity and intellectual property of forum participants.

We did contact with one stroke survivor who was a forum user between 2004 and 2011 in the context of a previous study (De Simoni et al, 2016). He has been supportive of our approach of reporting users' quotes, and even provided a positive statement about the research for the linked press releases.

3. The discussion shows that the findings are in line with previous findings, but there should be more focus on the new and interesting findings in both the results section and the discussion. Interesting findings that I think are currently hidden: a) There are various comments on decision-making with or without health providers. These is interesting and I think would be a useful discussion point in relation to the literature on self-management, and I think there could be more focus on the contributions this study makes to this topic. For example, on p18 authors note that non-adherence sometimes resulted from patients not having time to keep going back to the GP for check-ups. There is also mention of the role of pharmacists in giving advice. Currently this is categorised under 'barriers' on p19, and this does not seem a good fit. If authors structured the findings as suggested above, this would just come under 'practical factors' and authors would have flexibility for a more nuanced explanation of how such factors might influence adherence. This could then be taken up in the discussion with consideration of how patients can be supported and empowered to make informed decisions.

Response: The re-organisation of the results has resulted in an improved visibility of these important themes. At page 21, line 534 we have added the subtheme 'Seeking advice from pharmacists on managing medications' and a section titled 'questioning prescribing practices' (see page 16, line 394). The authors have described the issues around shared-decision making and the use of GPs' advice about secondary prevention medications in a sister paper, just published in Family Practice (Izuka et al, 2017) and therefore have not explored it extensively in this manuscript.

We have included the following statement in the discussion section of the manuscript, see page 26, lines 654-659:

'This study highlights a couple of interesting findings. Survivors reported making decisions about taking or not secondary prevention medications sometimes independently from their GPs, despite considering GPs' support important. Collaborative decision making involving caregivers, clinicians or pharmacists may however empower stroke survivors to make better informed decisions about secondary prevention medications. Understanding how patients make decision about medications is important (Benson & Britten, 2002) and GPs may benefit from enhancing caregivers' role in the decision making process about medications.

b) Data on patients with communication impairments and their caregivers (p 18-19). This is an interesting finding and shows the benefit of using the chosen method of data gathering.

Response: Thanks for this suggestion. The following paragraph has been added to the discussion, see page 25, lines 620-622:

Given that patients with significant disabilities may not traditionally participate in health research, the online forum may represent a potentially important method of data collection through which patients' views may be heard through their caregivers.

c) Strategies for medication management. Currently this section beginning on p20 seems muddled. The heading 'storage devices for medication management' should be 'storage devices and strategies for medication management', as authors include data that is not about storage devices (e.g. making a note, using a chart). At the start of this section authors state that storage devices ensured the appropriate medication was taken, but this contradicts data presented in the previous section. Again, if this was presented under 'practical factors' it would be possible to discuss the fact that storage devices and strategies may or may not be helpful.

Response: The contradictory views expressed around the use of medicine storage devices are

interesting. While these devices are known to improve medication taking in patients, difficulties may persist particularly for those patients with more severe disabilities as a result of stroke. This adds to the importance of caregivers' role in ensuring adherence to secondary prevention medications. In the discussion on page 28, lines 715-716, we have mentioned the importance of appropriate use of tablets storage devices.

'...appropriate use of tablet storage devices, particularly for those patients with more severe disabilities as a result of stroke, can increase adherence and ultimately improve health outcomes.'

Minor comments:

1. Discussion of the theory is lacking. Some critical discussion of why it was chosen is needed. On p5 the theory is explained but this could be re-worded for clarity, especially the sentence ending 'as well as unintentional adherence' around line 44. Linked with major comment 1) above, it is not clear how the theory has influenced the authors have presented the data.

Response: We have provided further clarification and description of the PAPA framework used in the study on Page 5, lines 136-137 and page 6, lines 139 -143 of the Methods section in the manuscript. The results have been reorganised to reflect the PAPA framework, with themes split into practicalities and perceptions.

2. p8 lines 19-23. This is a bit confusing. Did the majority of participants have a stroke within 12 months of posting, or post on the forum within 12 months of having a stroke? I would also suggest using years and months for time since stroke rather than e.g. 0.8 years.

Response: We have clarified this point by stating that participants took part in the forum within 12 months of having suffered from a stroke (see page 9, lines 228-229). We have also changed the unit of time (years and months), see page 9, line 231.

3. p13 second quote needs re-wording - currently unclear

Response: The quote has now been re-written to improve clarity. See page 16, lines 408-410.

4. Use of apostrophes needs correcting throughout. This is my reason for stating that the standard of English is not currently suitable for publication in this journal.

Response: We have corrected the use of apostrophes throughout the manuscript.

Reviewer: 2

Reviewer Name: Nathalie Moise

Institution and Country: Columbia University Medical Center, USA Competing Interests: None declared

Jamison et al. provide an interesting qualitative analysis of factors affecting medication adherence amongst stroke survivors and their caregivers using data from an online health forum. Strengths include inclusion of caregivers as well as disabling patients, both under-assessed in the post-stroke adherence literature, use of online forums reducing self-serving biases and reducing bias introduced by focus group facilitators and a theoretical framework for data analyses. However, the article methodology, which relies on a query of an existing forum without using interview guides/repeat interviews, is a key flaw and may provide a biased viewpoint, though the authors reference 1 prior article that used a similar methodology published in this journal.

Introduction:

1. The authors provide a nice overview of the literature and gaps in viewpoints of patients with debilitating strokes. The authors might strengthen their argument by positing why/how their assessment of barriers and facilitators of adherence might differ from prior literature other than reducing "self-presentational bias", which would affect all adherence literature.

Response: We thank the reviewer for this observation that has prompted further reflections. We have mentioned in the Introduction that reactivity bias can also be reduced when using an online forum as source of data, see page 4, lines 102-103.

We are not sure whether self-serving biases can be reduced by analysing patients' views from an online forum and have opted not to mention this.

We have also included the following sentence in the Introduction, see page 4, lines 106-109:

Our analysis differs from previous adherence literature by assessing survivors and caregivers attitudes to medication adherence from a viewpoint that has not been previously explored. The online forum offers users the opportunity to discuss issues around medication that may be considered sensitive and which they may be less willing to address through traditional face to face approaches.

2. The authors seem to interchangeably use self-“presentational” and “representation” in the introduction.

Response: We thank the reviewer for spotting the inappropriate use of the term, and have corrected the text accordingly, see page 4, line 100 and page 5 line 115.

3. While using an anonymous online forum is a strength, authors should include whether any prior research has used this methodology in assessing barriers/facilitators to adherence, specifically.

Response: We have added the following sentence to the Introduction section, page 4, lines 103-105 of the manuscript:

In a recent investigation, De Simoni and colleagues used an online forum to explore adherence to inhaler treatment in asthma adolescents, gaining fresh insights on factors affecting adherence in this patients' group.

Methods

4. The authors should provide an argument for using this framework specifically as opposed to any others

Response: We have provided additional reasoning for the use of the PAPA framework approach. The authors have included the following sentence in the Methods section of the manuscript on page 5, lines 136-137 and page 6 lines 139 -143.

The PAPA framework was chosen as it is specifically designed to identify and classify factors affecting adherence to medications. Results have the potential to inform the development of behavioural interventions aimed at improving adherence and their subsequent evaluation according to causal pathways.

5. How were keywords predefined (literature search?). How were authors assured that these would provide barriers/facilitators to adherence, specifically?

Response: We have included explanation as to how the keywords were chosen in 'Procedure and participants' within the Methods section on page 7 lines 171-175:

These keywords were lay terms used by patients with stroke and their caregivers when talking about adherence to secondary prevention medications as emerged from the transcripts of a previous interview study by the authors (Jamison et al, 2016). The aim of the interviews was exploring stroke survivors' and caregivers' views around barriers and facilitators of adherence to secondary prevention medications in general practice.

6. It would be helpful to better understand how TalkStroke is utilized and why. Do all caregivers use this forum and for what purpose? Can they discuss any topic or are there themes or subgroups that guide discussions? Given that many stroke patients had debilitating symptoms, how were authors able to clearly distinguish between the patient and caregiver (who themselves might assist the patient in posting on the forum?).

Response: We have added further explanation as to how Talkstroke was used and why, see the statement in the 'Setting' section of the Methods on page 6, lines 155-162:

Forum users could discuss any topics, develop their own conversation threads and there was no

restriction on the subject discussed. Participants could read the subject of the thread being discussed and decide whether they wished to contribute. Differentiating survivors and caregivers was done by reading the text of the post: survivors talked in first person about themselves, while caregivers were talking about a stroke survivor in the third person, e.g. 'my father had a stroke'. Stroke survivors with severe disabilities were amongst the users of the forum. Caregivers could register as users independently from patients. 60% of users were in fact caregivers (De Simoni et al, 2016). We acknowledge that some caregivers could have assisted patients in writing their posts, though we do not have data to confirm this hypothesis.

7. While open ended online forums allow for analyses of unprompted concerns around adherence, there are some methodological flaws that exist without interview guides and pre-specified questions/themes (e.g., identifying concerns related to adherence, specifically). Searching for "blister" provides very different implications from specifically elucidating (even on an online forum) barriers/facilitators to adherence in stroke patients, and contributes to measurement/coding bias. This method does not allow the authors to adhere to many of important COREQ items. How did the authors seek to mitigate this flaw? Is there literature on performing rigorous qualitative online forum analyses?

Response: We agree this approach has its own limitations, which, as discussed at page 25, lines 626-628 includes the fact that interrogating the dataset with different keywords might have revealed additional themes or revealed issues related to medications in general rather than specifically secondary prevention ones.

Despite the lack of possibility of asking questions to clarify themes, the fact that forum users could read about the subject of discussions and reply to the topic they felt they had something to say about, generated discussions between survivors and caregivers that offered great insights on barriers and facilitators to adherence, which may be outside the reach of interviews. Similarly, an investigation comparing an online forum with qualitative interviews as data sources in patients with cancer concluded that Internet forums provide useful data for qualitative health research.(Seale et al, 2010) The authors acknowledge that some COREQ items were not applicable and accept that this could be a limitation.

We added the following paragraph, see page 8, lines 204-207:

While we were unable to ask questions to clarify themes, users could participate in forum discussions they were interested in, offering insights on barriers and facilitators to adherence that may be beyond the reach of interviews. A previous investigation comparing an online forum with qualitative interviews concluded that the forum could provide useful data for qualitative health research. 24

8. The authors should provide a rationale for double coding 50% of the posts.

The authors should provide a kappa score

Response: We have provided further justification for the double- coding. Coding was repeated independently by a second author on a random sample of 50% of posts, with a kappa score of 80%. Therefore it was deemed not necessary to double code the whole dataset.

See the paragraph in the Methods on page 8 lines 192-194:

Throughout the process the authors checked the coding structure obtained to ensure a high level of agreement in coding was maintained. Once completed, coding were compared and the intercoder reliability was measured. The kappa score was 80%.

Results

The authors provide comprehensive data on characteristics/demographics of the participants, despite the fact that this is an online forum. I found the data and results to be quite rich and interesting.

10. It would be helpful to quantify the number of individuals in each subgroup/theme.

Response: We did not quantify the number of study participants within each theme. Traditionally,

qualitative research does not focus on numbers when reporting themes and the absence of counting is often considered the hallmark of qualitative research. Although multiple instances of a theme may be identified across a data set, the number of times this theme occurs does not make it more significant (Braun & Clarke, 2006).

11. The results were long and confusing. It would be helpful to better organize the results [e.g., changing “concerns” to “necessity concerns”; adding survivor headings to caregiver headings, placing all barriers together followed by facilitators, using similar fonts to specify themes, using more subheadings to differentiate different points, streamlining/shortening the caregiver results particularly since exact quotes are not utilized.

Response: The results section has been restructured considerably based on the recommendations of both reviewers. See pages 10-23.

As suggested, additional subheadings have been used to differentiate themes. The authors have also streamlined the results section by reducing the number of example quotes.

12. It might be helpful to choose 1-2 examples of each subthemes (e.g., on pgs 10-11 there are too many examples of necessity concerns for “burden and management of medication side effects” unless some of these should be sub-themes)] Unclear how authors were able to attribute these factors as adherence concerns. Was there high agreement among authors? Were posts excluded if not specifically pertaining to affecting adherence and how as this decided?

Response: we have followed the reviewer’s suggestion and have chosen 1-2 examples for each sub-theme. We have also added subheadings to clarify how the factors presented acted as barriers to adherence. About agreement among authors, we have added in the Methods that ‘throughout the process the authors checked the coding structure obtained to ensure a high level of agreement in coding was maintained. Once completed coding was compared and the intercoder reliability was measured. The kappa score was 80%.’, see page 8, lines 192-194.

13. It’s unclear why the authors chose to focus on adherence to statin/aspirin as opposed to also focusing on blood pressure agents for the purposes of secondary prevention.

Response: the keyword search included all secondary prevention medications, not only statins and antiplatelets/anticoagulants, see page 7, lines 166-170.

‘Terms related to secondary prevention medications were selected (e.g. Amlodipine, statin, warfarin, ramipril), including misspellings (e.g. Aspirin, simvastin), brand names (e.g. Lipitor, Plavix) and drug categories (e.g. statin, diuretics, blood pressure medicines etc.). Posts including any secondary prevention medication term were identified.’

Although most posts were about cholesterol lowering and antiplatelets and anticoagulants agents, there were a few example of barriers and facilitators to adherence to blood pressure medications, see page 11, lines 288-291; page 13, lines 338-342; page 17, lines 440-442; page 19, lines 472-474.

14. Not all of the caregiver/survivor themes seemed appropriate under the same headings (“difficulty taking medications” has different implications than “difficulty administering medications” as an overburdened caregiver)

Response: Both ‘difficulties taking medications’ and ‘difficulties administering medications’ represent potential practical barriers to patients’ adherence. We have changed the heading of the theme on page 19, line 493 to read “Problems associated with taking tablets”.

Discussion

Very nice overview of results and comprehensive review of limitations.

15. Minor edit on page 23: “older people know(n)”

Response: Thank you for spotting this error. This sentence now reads: With older people known to....See page 26, line 650-651.

16. While the authors highlight the ways in which their findings are in line with prior literature, it would be helpful to note the ways in which their findings also add to the literature. Did they find any differences from prior literature?

Response: We have added a paragraph on page 28, lines 700-705 of the manuscript, indicating how the findings add to the current literature.

These findings add to current literature by providing an assessment of adherence from users of an online forum. There has been little research on this approach to data collection conducted to date. The study identifies adherence concerns of a younger stroke population who may be less likely to be represented in research studies and whose attitudes to medication may be less well known. Findings add to the literature and shed light on dynamic interactions between the survivor, caregiver and health professionals and the extent to which this influences medication adherence in this patient group.

17. The authors might also highlight the ways in which caregiver/survivor beliefs differed, if at all.

Response: We thank the reviewer for this suggestion. We have now included in the discussion the following paragraph. See page 25, lines 640-645:

'Beliefs about secondary prevention medications differed at times between survivors and caregivers. Some stroke survivors decided to stop medications because of intolerable side effects, despite their caregivers' believing optimal adherence was important to prevent stroke recurrences. In the context of medication side effects, caregivers believed in their role as patients' advocates with healthcare professionals (including GPs and pharmacist) and often discussed and sought advice from other users in the forum on the matter'.

18. While the future research section is strong, it would be helpful to better elucidate the ways in which these findings improve implications ("implications for clinical research" section), as their proposed implications are similar to prior literature.

Response: We have added the following paragraph to the 'Implications for clinical practice' section on page 29, lines 725-732 of the manuscript:

These findings provide new insight to clinicians about younger stroke survivors' concerns and the struggles caregivers might face in their role as patients' advocates. Awareness of these factors will improve consultations about secondary prevention medication with both younger survivors and stroke survivors' caregivers. Stroke survivors with severe disabilities and their caregivers experience significant practical barriers to adherence. Greater focus on such practicalities by healthcare professionals would be beneficial.

This study highlights caregivers' unique position in overseeing patients' medications. Exploring the stroke survivor-caregiver dynamic can shed light on potential barriers to adherence to secondary prevention medication and ways to address them, eventually improving patients' outcomes.